# ITGAM rs1143679 Variant in Systemic Lupus Erythematosus Is Associated with Increased Serum Calcification Propensity

**DOI:** 10.3390/genes14051105

**Published:** 2023-05-18

**Authors:** Matthieu Halfon, Li Zhang, Driss Ehirchiou, Vishnuprabu Durairaj Pandian, Suzan Dahdal, Uyen Huynh-Do, Andreas Pasch, Camillo Ribi, Nathalie Busso

**Affiliations:** 1Transplantation Center, Lausanne University Hospital, 1010 Lausanne, Switzerland; matthieu.halfon@chuv.ch; 2Department of Physiology, Center for Vascular and Inflammatory Diseases, University of Maryland School of Medicine, Baltimore, MD 21201, USA; lizhang@som.umaryland.edu (L.Z.); vpandian@som.umaryland.edu (V.D.P.); 3Service of Rheumatology, Department of Musculoskeletal Medicine, Lausanne University Hospital, University of Lausanne, 1010 Lausanne, Switzerland; driss.ehirchiou@chuv.ch; 4Division of Nephrology and Hypertension, University Hospital Bern Inselspital, 3010 Bern, Switzerland; suzan.dahdal@insel.ch (S.D.); uyen.huynh-do@insel.ch (U.H.-D.); 5Swiss Systemic Lupus Erythematosus Cohort Study; camillo.ribi@chuv.ch; 6Department of Physiology and Pathophysiology, Linz University, 4040 Linz, Austria; andreas.pasch@hin.ch; 7Division of Immunology and Allergy, Department of Medicine, Lausanne University Hospital, University of Lausanne, 1010 Lausanne, Switzerland

**Keywords:** CD11B, lupus, calcification propensity, complement, ITGAM

## Abstract

Objectives: CD11B/ITGAM (Integrin Subunit α M) mediates the adhesion of monocytes, macrophages, and granulocytes and promotes the phagocytosis of complement-coated particles. Variants of the ITGAM gene are candidates for genetic susceptibility to systemic lupus erythematosus (SLE). SNP rs1143679 (R77H) of CD11B particularly increases the risk of developing SLE. Deficiency of CD11B is linked to premature extra-osseous calcification, as seen in the cartilage of animals with osteoarthritis. Serum calcification propensity measured by the T50 test is a surrogate marker for systemic calcification and reflects increased cardiovascular (CV) risk. We aimed to assess whether the CD11B R77H gene variant is associated with a higher serum calcification propensity (i.e., a lower T50 value) in SLE patients compared to the wild-type allele (WT). Methods: Cross-sectional study incorporating adults with SLE genotyped for the CD11B variant R77H and assessed for serum calcification propensity with the T50 method. Participants were included in a multicenter trans-disciplinary cohort and fulfilled the 1997 revised American College of Rheumatology (ACR) criteria for SLE. We used descriptive statistics for comparing baseline characteristics and sequential T50 measurements in subjects with the R77H variant vs. WT CD11B. Results: Of the 167 patients, 108 (65%) were G/G (WT), 53 (32%) were G/A heterozygous, and 6 (3%) were A/A homozygous for the R77H variant. A/A patients cumulated more ACR criteria upon inclusion (7 ± 2 vs. 5 ± 1 in G/G and G/A; *p* = 0.02). There were no differences between the groups in terms of global disease activity, kidney involvement, and chronic renal failure. Complement C3 levels were lower in A/A individuals compared to others (0.6 ± 0.08 vs. 0.9 ± 0.25 g/L; *p* = 0.02). Baseline T50 did not differ between the groups (A/A 278 ± 42′ vs. 297 ± 50′ in G/G and G/A; *p* = 0.28). Considering all sequential T50 test results, serum calcification propensity was significantly increased in A/A individuals compared to others (253 ± 50 vs. 290 ± 54; *p* = 0.008). Conclusions: SLE patients with homozygosity for the R77H variant and repeated T50 assessment displayed an increased serum calcification propensity (i.e., a lower T50) and lower C3 levels compared to heterozygous and WT CD11B, without differing with respect to global disease activity and kidney involvement. This suggests an increased CV risk in SLE patients homozygous for the R77H variant of CD11B.

## 1. Introduction

Systemic lupus erythematosus (SLE) is a chronic autoimmune disease affecting various organ systems [1]. SLE patients are at increased risk of cardiac disease due to autoimmune arteriosclerosis [2]. Serum calcification propensity measured by the T50 test is a surrogate marker for systemic calcification in a given individual. The T50 value is the time necessary to convert primary calciprotein particles (CPP) in the patient’s serum to their secondary form. For SLE patients, T50 values have been shown to be associated with disease activity but have also proven to be predictors of cardiovascular diseases [3]. Therefore, T50 values may be an important tool linking inflammation, disease activity, and cardiovascular outcome in SLE patients.

SLE arises in genetically susceptible individuals under the influence of endogenous and environmental factors [1]. Among the candidate genes involved in the genetic susceptibility for SLE features is ITGAM (CD11B, α_M_), which codes for the α subunit of the complement receptor 3 (CR3, Mac-1, CD11B/CD18, or α_M_β_2_). CD11B is expressed in a wide variety of immune cells, such as macrophages, neutrophils, dendritic cells, and a small percentage of lymphocytes [4]. Indeed, CD11B is implicated in the recruitment and activation of neutrophils but also in the clearance of immune complexes [4].

Genome-wide association studies have shown that single-nucleotide polymorphisms (SNP) of CD11B affect the structure and function of this protein [5]. Individuals with the SNP r1143679 G/A, which changes from arginine (CGC) to histidine (CAC) at position 77 (R77H) of CD11B, are at an increased risk of developing SLE [6]. Dysfunction of CD11B was recently implicated in the mechanism of cartilage calcification due to enhanced secretion of interleukin (IL)-6 [7]. Moreover, ITGAM gene expression has been implicated as a candidate gene for better stratified patients with an acute myocardial infarction [8].

An increase in calcifications due to ITGAM dysfunction might play a role in the increased risk of cardiovascular diseases observed among SLE patients. In this study, we hypothesized that SLE patients with the CD11B (R77H) SNP compared to those with the common allele had a higher propensity for developing systemic calcifications measured using the T50 score.

## 2. Methods

This cross-sectional study incorporated subjects that had been included in the Swiss SLE Cohort Study (SSCS) between April 2007 and December 2013 [9]. SSCS was a multi-centric and transdisciplinary observational study that was approved by the ethics committee (CER-VD) (BASEC-2017-01434) [9]. All participants in SSCS gave their written informed consent. Subjects included in this study were aged ≥18 years, fulfilled the 1997 revised ACR criteria and/or the 2012 SLICC criteria for SLE, and had been assessed for their propensity to develop calcification. Extracted DNA and serum samples were obtained, and subjects were genotyped for CD11B gene variants. Data at baseline included demographic characteristics, disease manifestations and comorbidity, disease activity scores, and laboratory values. Longitudinal measurements included T50 measurement. Global disease activity was assessed via the SELENA-SLEDAI (Safety of Estrogens in Lupus Erythematosus National Assessment—SLE Disease Activity Index) [9]. Organ damage was assessed using the SDI (Systemic Lupus International Collaborative Clinics/American College of Rheumatology SLE Damage Index) [9].

### 2.1. Biological Sample

Serum samples were drawn from a peripheral vein in vacutainer tubes. After 30 to 60 min, samples were centrifuged at 4000 rpm (corresponding to 2600 G) for 15 min at ambient temperature and the extracted serum was stored in aliquots and frozen at −80 °C until further use. Serum calcification propensity was assessed according to the T50 test. In brief, the T50 test is an in vitro test that quantifies the calcification inhibition inherent in blood by challenging the patient’s serum with supersaturated calcium and phosphate solutions.

This leads to the instantaneous formation of CPP. The timing of the spontaneous transformation of these particles into secondary CPP depends on the specific composition of the serum and, specifically, the concentrations and interplay of well-established calcification-inhibiting factors [10]. Serum samples were challenged with highly concentrated calcium and phosphate solutions to induce the formation of primary CPP. The ripening and spontaneous transformation from primary to secondary CPP was then monitored in a time-resolved manner using a standard nephelometer (Nephelostar; BMG Labtech, Ortenberg, Germany). The results of these measurements were used to calculate the one-half maximal transition time (T50) in minutes.

A shorter transformation time indicates a more rapid precipitation of calcium and phosphate in the presence of serum.

All serum samples were measured under blinded conditions at the Department of Nephrology, Hypertension and Clinical Pharmacology, University Hospital Bern, Bern, Switzerland. Samples were measured in triplicate. T50 is stable when samples were stored at -80 ˚C throughout as has been demonstrated in previous studies [11].

### 2.2. Genotyping of ITGAM SNPs

ITGAM SNPs of 187 clinical samples were genotyped using the TaqMan SNP genotyping assay system (Applied Biosystems, Waltham, USA) in a 96-well MicroAmp Optical plate. The PCR mixture used consisted of 10ng genomic DNA, 12.5 μL of TaqPath ProAmp Master Mix, and 1.25 μL of validated TaqMan SNP Genotyping primer mix (Assay IDs rs1143683:C_11388882_20; rs1143679:C_2847895_1). The total volume was 25 μL. Allelic discrimination with endpoint detection of fluorescence was performed in QuantStudio-3 (Applied Biosystems, USA). Polymerase chain reaction (PCR) was initiated using the manufacturer’s recommended protocol, whose steps are as follows: pre-read for 30 s at 60 °C, initial denaturation for 5 min at 95 °C, and 40 cycles of 15 s at 95 °C and 60 s at 60 °C, followed by post-read for 30 s at 60 °C. Non-template control and positive control (known homozygous variants) were routinely added in each reaction plate. SNP genotypes were analyzed using QuantStudio Design & Analysis software v1.5.1 (Applied Biosystems, USA).

The SNP genotype results (all homozygous and one randomly selected heterozygous) were confirmed via DNA sequencing. The CD11B exon 3 encompassing SNP rs1143679 was amplified from genomic DNA using the forward 5′-CTCTGTTCCCACTTCTCCCC-3′ and reverse 5′-AGGCAGAGGAGAGGGTACG-3′ primers. The PCR was initiated as follows: 95 °C for 5 min, followed by 35 cycles of 95 °C for 30 s, 60 °C for 30 s, and 72 °C for 30 s. The PCR-amplified product was cleaned using NucleoSpin PCR clean-up spin-column kit (Macherey-Nagel GmbH, Düren, Germany). DNA sequencing was carried out on a 3730XL DNA Analyzer (Applied Biosystems, USA), and the accuracy of all the TaqMan SNP genotype results was confirmed.

### 2.3. Statistical Analysis

Data are presented as absolute numbers with percentages for categorical data, as means ± standard deviation (SD) for continuous variables, and, if not normally distributed, as medians with interquartile range (IQR). Between group analyses was performed using Chi-squared test and Fisher’s exact test for categorical data and Student’s T-test or Mann–Whitney U test for continuous data. Multigroup analysis was performed using the Kruskal–Wallis test. Given the descriptive nature of the study, we did not adjust for multiple comparisons. *p* < 0.05 was considered statistically significant. A longitudinal mixed model test for variance was used to account for intra-subject T50 value variance by comparing all sequential measurements for each subject. For between group comparison, we pooled all values from sequential T50 test measurements.

## 3. Results

Baseline characteristics of the SLE patients are shown in Table 1.

Of the 167 individuals included, 144 (86%) were females and 129 (78%) were of European descent. The mean age upon SLE diagnosis was 34 years, and the age at first assessment was 43 years. SNP analysis for R77H showed 108 (65%) individuals with G/G (wild-type) homozygous, 53 (32%) with G/A heterozygous, and 6 (3%) with A/A homozygous. Between-groups clinical parameters at inclusion are shown in Table 2.

The A/A homozygotes had more 1997 revised American College of Rheumatology (ACR) criteria for SLE at inclusion (7 ± 2 vs. 5 ± 1; *p* = 0.02). SLE psychosis was rare, but it was more frequently observed in A/A patients ((32%); *p* = 0.01). Baseline T50 levels did not differ between groups (G/G and G/A: 297 ± 50 min; A/A: 278 ± 42 min; *p* = 0.28). Sequential T50 measurements displayed low intra-subject variation (*p* = 0.1). Pool T50 measurements were also analyzed: of the 167 patients, 104 had several T50 measurements, and the mean number of the T50 measurement for patients was 2 ± 1. For patients will multiple measurements, the mean time between the first and last measurement was 27 ± 15 months. Pooled T50 measurements were significantly shorter in subjects homozygous for R77H SNP A/A (G/G and G/A: 290 ± 54 min vs. A/A: 253 ± 50 min; *p* = 0.008), indicating a higher propensity for calcification (Figure 1).

As there was no statistical difference between groups for the baseline T50 value but there was one for the pooled T50 value, we analyzed the difference between baseline SELENA-SLEDAI and the last available SELENA-SLEDAI for G/G and G/A as well as A/A to see if there was a time difference between groups. For G/G and G/A, there was a nearly significant trend toward less aggressive disease over time with a mean reduction of −1.3 + −9.2 (95%CI −3.4 to 0.2) (*p* = 0.07) of the SELENA-SLEDAI; for A/A, there was a non-significant increase of + 2.7 + −4.1 (95%CI 7.7 to 13.0) (*p* = 0.38). Finally, to better analyze the allele effect, we also computed the pooled T50 measurements between wildtype homozygous (G/G) vs. genotypes heterozygous and homozygous for R77 SNP (G/A and A/A): the pooled T50 values were 287 ± 60 min for G/G and 290 ± 45 min for G/A and A/A (*p* = 0.65), respectively.

The complement C3 levels upon assessment were lower in the A/A subjects compared to the G/G and G/A subjects (0.6 ± 0.08 g/L vs. 0.9 ± 0.25 g/L; *p* = 0.03). When all three groups were considered, the C3 levels were still significantly lower in the A/A group compared to the other groups (G/G: 0.9 ± 0.25 g/L, G/A: 0.9 ± 0.44 g/L, and A/A: 0.6 ± 0.08 g/L; *p* = 0.05).

## 4. Discussion

In our study, subjects homozygous for the CD11B variant R77H A/A had more severe SLE and displayed lower C3 levels and an increased serum calcification propensity measured via the T50 test compared to those heterozygous or the wild-type.

Serum calcification propensity measured via the T50 test is associated with the cardiac outcomes of patients with SLE, for which patients with low T50 values have a higher risk than patients with high T50 values [3]. Our results showed that patients homozygous for R77H SNP have a lower T50 value than wild-type and heterozygous patients. However, there was not a difference in serum calcium propensity when wild-type homozygous individuals were compared to individuals heterozygous and homozygous for R77H SNP. This could suggest that R77H SNP could function as a recessive variant and that one wild-type allele is enough to ensure the physiological function of the gene.

Interestingly, CD11B-deficient mice have been shown to display a propensity to develop cartilage calcification, which is partly due to the increased secretion of IL-6 by CD11B-deficient chondrocytes [7]. Moreover, inflammation, for which IL-6 is a key mediator, is a major driving mechanism of accelerated arteriosclerosis [2]. Indeed, patients with low T50 levels have high concentrations of inflammatory markers [12]. Our results suggest that CD11B dysfunction might also promote systemic calcification, leading to an increased risk of cardiovascular diseases.

The loss of function of CD11B is associated with the impairment of the clearance of immune complexes by neutrophils and macrophages [13]. In SLE, this leads to more immune complexes being deposited in tissues, resulting in organ damage, which generally occurs via complement activation. We found that R77H was associated with significantly lower C3 levels but not C4 levels, suggesting an effect mainly on the alternative pathway of the complement. Of note, C3 levels seem to correlate more with disease activity than those of C4 [14]. A dysfunction of CD11B could affect neutrophils’ ability to inhibit the alternative pathway as they interact with CR3, which consists of CD11B and CD18 [15]. This could lead to more activation of the alternative pathway promoting more complement deposition [15], thus suggesting a more severe phenotype in homozygous patients. Indeed, our cohort of homozygous patients cumulated more SLE features than the wild-type patients, particularly with respect to neuropsychiatric SLE, which is associated with worse outcomes [16]. This finding is in line with the findings presented in previous studies, as T50 levels are inversely correlated with disease activity in SLE cases [3]. Interestingly, C3 levels have been shown to correlate with ABI and the angiographic parameters of atherosclerosis but not with the severity of calcification measured using an arterial calcification score [17]. However, serum calcium propensity measured using the T50 test has been proven to be correlated with vascular calcification [18]. Therefore, dysfunction in ITGAM might lead to increased calcification via complement-independent pathways.

Finally, in patients with a wild-type common allele and patients heterozygous for R77H SNP, there was a trend toward less aggressive disease over time compared to the patients that were homozygous for R77H SNP, who either displayed no change or an increase in global disease activity over time. Although the limited number of individuals in our study does not allow us to draw firm conclusions, this observation is in line with R77H reportedly being associated with a more severe disease course [19].

A significant limitation of our study was that our cohort mostly consisted of Caucasian patients and thus insufficiently accounted for ethnicity, which is known to play an important role in genetic association studies. Moreover, few patients were homozygotes; therefore, the findings of our pilot study should be replicated in a larger cohort.

## 5. Conclusions

In this pilot study, we found an increased serum calcification propensity (i.e., lower T50 value) in SLE patients affected by the variant for R77H SNP, which may be a direct result of CD11B dysfunction.

CD11B dysfunction is linked to more severe SLE, which presumably occurs via an increase in inflammation and alternative complement activation. This may translate into accelerated arteriosclerosis.

## Figures and Tables

**Figure 1 genes-14-01105-f001:**
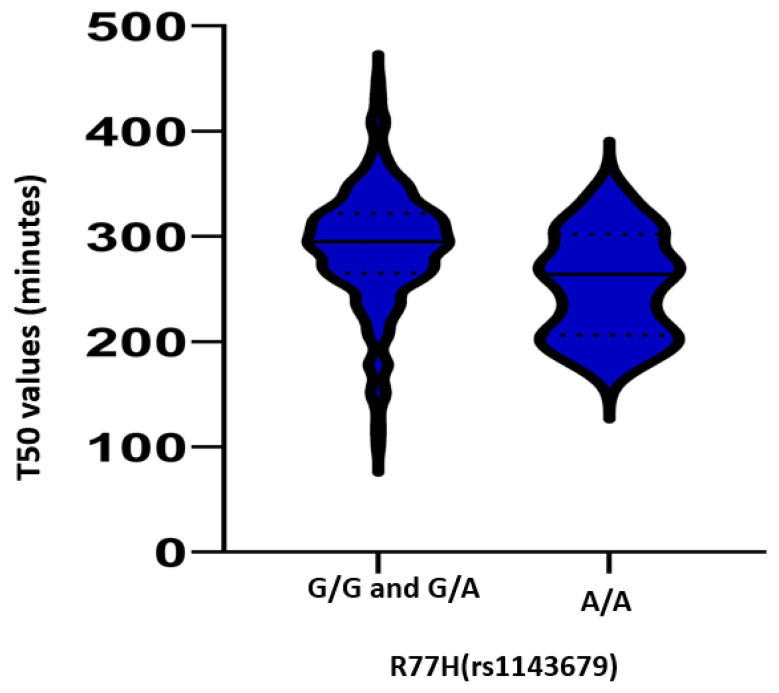
Serum calcium propensity measured using the T50 test in 356 assessments of 167 patients with systemic lupus erythematosus according to their CD11B R77H genotype (R77H G/G and G/A vs. R77H A/A). Solid line represents the median value, and dashed lines represent the 25th and 75th quartiles = 0.008.

**Table 1 genes-14-01105-t001:** Baseline characteristics of 167 patients with SLE according to their R77H genotype.

Variable	All Patients N = 167	R77H (G/G and G/A) N = 161	R77H (A/A) N = 6	*p*
Women, no. (%)	144 (86)	138 (85)	6 (100)	NS
Age at SLE diagnosis, mean ± SD, years	34 ± 15	34 ± 14	29 ± 9	NS
Number of ACR criteria at diagnostic	5 ± 1	5 ± 1	7 ± 2	0.02
Chronic renal failure, no. (%)	12 (7)	11 (7)	1 (16)	NS
Malar rash, no. (%)	68 (40)	67 (41)	1 (16)	NS
Discoid rash, no. (%)	31 (19)	28 (17)	3 (50)	0.04
Photosensitivity, no. (%)	77 (46)	74 (45)	3 (50)	NS
Nasopharyngeal ulcers, no. (%)	36 (22)	35 (21)	1 (16)	NS
Arthritis, no. (%)	137 (83)	131 (81)	6 (100)	NS
Pleuritis, no. (%)	39 (23)	34 (21)	5 (83)	0.001
Pericarditis, no. (%)	37 (22)	32 (19)	2 (32)	NS
Renal disorder, no. (%)	74 (44)	70 (43)	4 (64)	NS
Seizures, no. (%)	14 (8)	13 (8)	1 (16)	NS
Psychosis, no. (%)	8 (5)	6 (4)	2 (32)	0.001
Hematologic disorder, no. (%)	106 (64)	100 (62)	6 (100)	0.06
Anti-dsDNA antibodies positive, no. (%)	116 (70)	111 (69)	5 (83)	NS
Anti-Sm antibodies positive, no. (%)	38 (22)	35 (21)	3 (50)	NS
Anti-phospholipid antibodies positive, no. (%)	80 (48)	76 (47)	4 (64)	NS
ANA positive, no. (%)	162 (97%)	157 (93)	5 (83)	NS

**Table 2 genes-14-01105-t002:** Clinical and biological parameters of 167 patients with systemic lupus erythematosus at the time of inclusion.

Parameter, Mean ± SD	All Patients = 167	R77H (G/G and G/A) N = 161	R77H (A/A) N = 6	*p*
SLEDAI	5 ± 5	5 ± 5	5 ± 4	NS
ESR, mm/1st h	22 ± 21	22 ± 22	16 ± 8	NS
Hemoglobin, g/L	129 ± 15	130 ± 15	121 ± 14	NS
Leucocytes, G/L	5.9 ± 2.2	5.9 ± 2.1	5.9 ± 2.6	NS
Lymphocyte, G/L	1.3 ± 0.6	1.3 ± 0.6	1.53 ± 0.9	NS
Thrombocytes, G/L	223 ± 84	233 ± 84	233 ± 47	NS
Creatinine, µmol/L	73 ± 28	73 ± 29	79 ± 30	NS
C3, g/L	0.9 ± 0.25	0.9 ± 0.25	0.6 ± 0.08	0.03
C4, g/L	0.16 ± 0.8	0.15 ± 0.08	0.16 ± 0.13	NS
SLICC/ACR SDI	0.9 ± 1.5	0.9 ± 1.5	0.8 ± 1.3	NS

## Data Availability

The anonymized data are available upon reasonable request for a collaborative research project. Please contact the corresponding author for all requests regarding data sharing.

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
