# Peer review of "ITGAM rs1143679 Variant in Systemic Lupus Erythematosus Is Associated with Increased Serum Calcification Propensity"

_genes, 2023, doi:10.3390/genes14051105_

Round 1
Reviewer 1 Report
The authors measured the ability of serum to inhibit calcification in SLE patients genotyped for ITGAM polymorphism. Of the 167 patients 108 were wild type homozygous, 53 heterozygous and 6 homozygous for the aberrant CR3 allele. They found decreased calcification inhibitory activity in A/A homozygous patients.
Major remarks
The key message of the paper is about a calcification propensity measurement, yet the method is not described at all, only a reference is made to a previous publication. What are the measurement units of T50? What is the correct name of the measurement: calcification propensity or calcification inhibition (lower values reflect higher calcification propensity) or serum calcium propensity, as the legend of Figure 1 says?
Obviously 6 patients are not necessarily representative of a genotype effect. The ITGAM alleles can manifest in heterozygous form, that is the G/A genotype may also influence the examined serum property. Did the authors evaluate G/G versus G/A and A/A, in addition to the G/G and G/A versus A/A comparison shown in the paper? The fact that longitudinal measurements are stable does not strengthen the statement about the A/A allele effect but rather confirms that in these particular 6 patients T50 values are lower.
Did this study confirm that in reference 3, namely that T50 level is inversely correlated with SLE disease activity?
Minor remarks
Line 74 : Which ethics committee? ; Sentence starts with non-capital letter.
Line 86 : incomplete sentence
Figure 1 : What is the point in displaying sample IDs for the outliers? A box&whisker display for 6 samples not very informative, a bee-swarm plot, perhaps with color ID for the samples might give a better impression about the distribution of T50 values.
Overall, a deeper and an alternative allele effect analysis and better visualization of results might make up for the lack of sufficient sample number in the A/A group.
Reviewer 2 Report
The paper by Halfon et al. “ITGAM rs1143679 variant in systemic lupus erythematosus is associated with increased serum calcification propensity” describes a potentially interesting study aiming to evaluate the calcification potential of serum from LES patients. Data were correlated to serum C3 values and to the occurrence of the SNP R77H in the ITGAM gene.
Significant changes were observed in patients homozygous for the R77H variant compared to carriers and WT patients. Although the number of homozygous patients is low, results, if conformed in a larger cohort, can be of interest. As stated by Authors this is a pilot study, however, there is one aspect that must be clarified by Authors:
- Line 80: Authors mention that they performed “longitudinal measurements”. However, it is not mentioned after how many months/years after baseline measurements were the analyses repeated. Did all patients undergo repeated sampling? Looking at the number (356 assessments for 167 patients), it means that patients were evaluated more than twice. Is the second time point equal for all patients? These data must be provided. This is a crucial aspect, since baseline values were not different among groups, and changes were detected only when measures were pooled within groups. Is this the effect of the disease worsening in the homozygous group of patients? Authors should clearly comment on these data, since if there are no differences at baseline and therefore it seems that T50 cannot be used as a marker per se, but with a prognostic value only in association with the R77H SNP.
Minor comments
- Line 85: check the appropriateness of the title. Do you mean: Biological sampling or Biochemical analyses? See also line 86 that is not clear.
- Authors should comment on findings in other studies indicating that “C3 serum levels are associated with ABI and angiographic parameters of atherosclerosis, but do not relate to the severity of calcification “ (Int. Angiol. 2014:33:35-41)
Round 2
Reviewer 1 Report
The authors adequately addressed my concerns.
Please run a grammar check to correct some minor mistakes in the revised, red text (lines 154, 174).
Author Response
Please run a grammar check to correct some minor mistakes in the revised, red text (lines 154, 174).
The grammatical mistakes have been corrected.
Reviewer 2 Report
The manuscript has been adequately revised
Minor typing errors, e.g. :
line 153: minutes' instead of minutes
line 154: analysis instead of analysed?
line 156: - instead of .
........
Author Response
the manuscript has been adequately revised
we are pleased to have been able to answer the reviewer comments.
Comments on the Quality of English Language
Minor typing errors, e.g. :
line 153: minutes' instead of minutes
line 154: analysis instead of analysed?
line 156: - instead of .
The typing errors have been corrected.